# Classification Tree to Analyze Factors Connected with Post Operative Complications of Cataract Surgery in a Teaching Hospital

**DOI:** 10.3390/jcm10225399

**Published:** 2021-11-19

**Authors:** Michele Lanza, Robert Koprowski, Rosa Boccia, Adriano Ruggiero, Luigi De Rosa, Antonia Tortori, Sławomir Wilczyński, Paolo Melillo, Sandro Sbordone, Francesca Simonelli

**Affiliations:** 1Multidisciplinary Department of Medical, Surgical and Dental Sciences, University of Campania Luigi Vanvitelli, 80100 Napoli, Italy; rosa_boccia111@hotmail.com (R.B.); adriano.ruggiero@me.com (A.R.); luigidrderosa@gmail.com (L.D.R.); antonia.tortori@gmail.com (A.T.); paolo.melillo@unicampania.it (P.M.); sandro.omar65@gmail.com (S.S.); francesca.simonelli@unicampania.it (F.S.); 2Institute of Biomedical Engineering, Faculty of Science and Technology, University of Silesia in Katowice, Bedzińska 39, 41-200 Sosnowiec, Poland; robert.koprowski@us.edu.pl; 3Department of Basic Biomedical Science, Faculty of Pharmaceutical Sciences in Sosnowiec, Medical University of Silesia, Będzińska Street 39, 41-200 Sosnowiec, Poland; swilczynski@sum.edu.pl

**Keywords:** cataract surgery, complications, artificial intelligence, risk factors

## Abstract

Background: Artificial intelligence (AI) is becoming ever more frequently applied in medicine and, consequently, also in ophthalmology to improve both the quality of work for physicians and the quality of care for patients. The aim of this study is to use AI, in particular classification tree, for the evaluation of both ocular and systemic features involved in the onset of complications due to cataract surgery in a teaching hospital. Methods: The charts of 1392 eyes of 1392 patients, with a mean age of 71.3 ± 8.2 years old, were reviewed to collect the ocular and systemic data before, during and after cataract surgery, including post-operative complications. All these data were processed by a classification tree algorithm, producing more than 260 million simulations, aiming to develop a predictive model. Results: Postoperative complications were observed in 168 patients. According to the AI analysis, the pre-operative characteristics involved in the insurgence of complications were: ocular comorbidities, lower visual acuity, higher astigmatism and intra-operative complications. Conclusions: Artificial intelligence application may be an interesting tool in the physician’s hands to develop customized algorithms that can, in advance, define the post-operative complication risk. This may help in improving both the quality and the outcomes of the surgery as well as in preventing patient dissatisfaction.

## 1. Introduction

Phacoemulsification, with concomitant intraocular lens (IOL) implant, is the current standard technique for cataract surgery and represents one of the most commonly performed surgical acts in the world [1]. Thanks to the continuous improvement of IOL design and materials this technique allows vision restoration with or without spectacle aid [2,3]. Thus, today, patients have very high expectations after cataract surgery, even though a low rate of intraoperative and post-operative complications still exists [1,4]. Patient dissatisfaction is an important factor to consider, especially in a university teaching hospital, where more complicated cases are usually managed and where residents may make more mistakes compared to trained surgeons. 

Identifying challenging cases and those most likely to develop post-operative complications may considerably help physicians in reducing the number of issues after surgery, in providing patients with a realistic expectation of surgical results and in improving the selection of surgical cases for residents and fellows. 

One of the main limitations of the previously published studies on this topic is the analysis selection: in most cases a multivariate correlation method was applied [5,6,7,8,9,10,11,12,13,14,15]. However, this approach has a reliability problem caused by the evaluation of numerically very different groups: in fact, the number of patients who develop postoperative complications is much lower than those who do not develop them. In addition, the number of patients with ocular comorbidities is higher than those without them. [16,17]. For this reason, an alternative analysis, such as artificial intelligence (AI), has been applied to cataract surgery [10] and other medical topics [18,19,20,21,22,23,24,25]. AI methods permit the acquisition of information about a specific working condition, and in turn propose different approaches to improve the quality and/or efficiency of a specific aspect of the situation evaluated. Although it is well known that AI analysis has certain limitations [26,27], it has been more widely and successfully applied in studies where classifications of non-analytical characteristics might be better adapted [28,29]. In particular, the potential advantages of an AI approach in ophthalmology appear to be very promising, as can be deduced by the increase in its applications [30,31]. In particular, among available methods, classification tree algorithms have been adopted, because they provide a classification model, i.e., “if … then” rules, which are easy to read and to interpret. This is crucial in medical applications [32], in which the physician is personally responsible for decision-making. Alternatively, other methods, such as neural networks, are black boxes and require further analysis to extract human intelligible knowledge.

The most commonly studied early post-operative complications are transient elevated intraocular pressure (IOP), corneal oedema, toxic anterior segment syndrome, intraocular lens (IOL) decentration or dislocation and endophthalmitis. Late post-operative complications include posterior capsule opacification (PCO), clinical macular oedema (CMO), pseudo-phakic bullous keratopathy (PBK), chronic uveitis, retinal detachment and late endophthalmitis [4]. Many studies have evaluated one or a few complication rates together with the factors involved [11,13,14,15], whereas this is the first study which detects and analyses all issues occurring after cataract surgery through AI application.

## 2. Materials and Methods

This retrospective study includes 1392 eyes of 1392 patients (mean age of 71.3 ± 8.2 years) who underwent cataract surgery from January 2018 to January 2020 in the Ophthalmology Unit of Università degli Studi della Campania “Luigi Vanvitelli”. The study followed the tenets of the Declaration of Helsinki and was approved by the local Ethics Committee. Informed consent was obtained.

If both eyes underwent surgery, only data related to the first cataract operation was included to avoid bias in the statistical analysis related to the inner correlation between pair organs. The exclusion criteria included eyes having previously undergone vitrectomy, planned cataract surgery with extracapsular extraction, or combined surgical procedures such as phacoemulsification associated with vitrectomy, trabeculectomy or corneal transplant.

All patient charts were analysed and all data from preoperative to post-operative follow-up were collected. Before undergoing surgery, all patients underwent a comprehensive ophthalmologic examination including evaluation of intra-ocular pressure (IOP), blood chemistry tests, electrocardiogram and an anesthesiologic visit. All patients underwent OCT (RTvue, Optuvue, Freemont, CA, USA) macular scan and IOLMaster 500 (Zeiss, Jena, Germany) evaluation. IOL was calculated to reach emmetropia or refraction planned with the patient, using SRK/T formula for eyes longer than 22 mm and Hoffer Q formula for those shorter than 22 mm. Phacoemulsification was performed with temporal approach and a 2.75 mm clear corneal incision, with Constellation Vision System with Ozil Torsional phaco tip (Alcon, Fort Worth, TX, USA). IOL models implanted in the study population were mostly SA60AT (686; 49.28%) and SN60WF (323; 23.20%) (Alcon, Fort Worth, Texas, USA). Almost half of the procedures, 575 eyes (41.31%) were performed with topical anesthesia, 524 (37.64%) had peri-bulbar anesthesia, 289 (20.76%) had sub-tenonian anesthesia and four had general anesthesia. 

Trained staff surgeons and 3rd or 4th year residents in ophthalmology performed surgery. In our residency program, cataract surgery training starts from the 3rd year, while 2nd year residents perform only surgical acts on ocular adnexa. A total of 1192 surgical (85.63%) procedures included in this study were performed by trained surgeons whereas 200 (14.37%) were performed by residents, the more complex cases being assigned to trained surgeons. Every staff surgeon has at least 10 years’ experience and their skills can be considered as comparable. 

Routine follow up was scheduled for 1 day, 1 week, 1 month and 3 months post-surgery. Additional evaluations were scheduled according to the evolution of the clinical scenario. Apart from 1 day after surgery, follow up visits included visual acuity testing, IOP measurement, anterior segment bio-microscopy and OCT, while at 1- and 3- month follow-up indirect ophthalmoscopy was also performed. Mean follow-up of the eyes included in the study was 6.67 ± 8.76 months (ranging from 3 to 14 months). General and clinical features collected in this study are summarized in Table 1 and Table 2. 

The complications detected in this study were the following: PCO (1), CMO (2), long lasting corneal oedema solved without surgery (3), PBK (4), transient elevated intraocular pressure (5), partial iris atrophy (6), IOL decentration and/or dislocation (7) and unsatisfactory visual recovery after cataract surgery (8). No infectious or inflammatory complications, retinal tears and/or retinal detachment were detected.

Artificial Intelligence was applied to process the following parameters: age, sex, laterality of the eye, systemic diseases, ocular diseases, best correct visual acuity before surgery (BCVA), spherical refraction of patients before cataract surgery (SP), cylinder refraction of patients before cataract surgery (CIL), refraction expressed as spherical equivalent (SE), intraocular pressure (IOP), implanted IOL power, cataract type (total, cortico-nuclear, sub-capsular and cortico-nuclear plus sub-capsular), axial length (AL), steeper keratometry (Kmax), flatter keratometry (Kmin), mean keratometry (MK), anterior chamber depth (ACD) measured from the epithelium, endothelial cell count, type of anesthesia, extra devices used during surgery, time of surgery, intraoperative complications and type of surgeon (trained or in training).

The normality of the distribution was checked with the Kolmogorov-Smirnov test, and the difference between uncorrected visual acuity (UCVA) and best corrected visual acuity (BCVA) before and after surgery was evaluated with the paired Student T test and with the mean square error (MSE).

### Classification Tree Application

The decision trees proposed in this article are machine learned [33]. Machine learning is an area of artificial intelligence dedicated to algorithms that improve automatically through the experience of medical research, i.e., exposure to data [33]. Machine learning algorithms build a mathematical model from sample data, called a training set, to forecast or make decisions without being explicitly programmed by humans for that purpose [33,34].

Currently, there are many methods of artificial intelligence (machine learning) that are used in various fields of medicine [18,19,20,21,22,23,24,25]. These methods have various advantages and disadvantages. In the case of artificial neural networks, support vector machines or principal component analysis (PCA) methods, it is not possible to create a clear system of division between classes: healthy or sick. In the case of neural networks or deep learning (for which we have too little data in this article), this is a black box [33,34]. In the case of PCA or support vector machines (SVM), as a result of the analytical form, there is no relationship between the two classes. One clear form in terms of visualization and the creation of an algorithm is the decision tree [33,34]. These not only have the ability to present the relationships between classes in the form of a graph, but also to visualize the limit values for individual features. An example is information where the first most important criterion is division into intraocular pressure which is greater or less than a given threshold (in the case of binary decision trees). This allows not only an easy understanding of the operation of such an algorithm by doctors, but also an attempt to interpret the dependencies and conditions created by the algorithm for individual analyzed features. No other method of artificial intelligence has this type of possibility or parameters [33,34].

The preliminary results of the analysis were carried out for all available data, i.e. 23 different features evaluated in 1392 eyes. These studies aimed to determine the best artificial intelligence method to apply in this case. The analysis covered artificial neural networks with back propagation errors, a support vector machine, decision trees and naive Bayes classifier. The accuracy (ACC) results are presented in Table 3.

For this reason (ease of visualization and best results) binary decision trees were used for classification. All data were split into training and test vector in three-quarters, and the remaining one quarter used ratios. Calculations, i.e., generating a binary decision tree, pruning the decision tree, checking the correctness of operation, were performed for each possible configuration of 28 features or 223 = 8,388,608 tests. Due to the low efficiency in the classification of a small number of data sets for a large number of features [23], the number of data combinations with a maximum of five features was limited, i.e., to 122,437 cases of decision trees. The simulations were made using a computer with an Intel Xeon 3.3 GHz processor, 12 GB RAM. The algorithm was written in the Matlab Version: R2016a, with toolboxes Signal Processing, Statistics and Machine Learning Toolbox Version 10.2 (R2016a). The simulation was performed by generating a decision tree for each of the possible parameter configurations. All simulations are limited to decision trees because they allow easy visualization and easy understanding of the division adopted, thanks to the algorithm that distinguishes individual complications.

Finally, for the purpose of comparison of a binary decision tree with previous studies, multivariate analysis was conducted using a conditional logistic regression model: each variable with a *p*-value < 0.05 was included in the multivariate analysis, whereas a stepwise approach was used to exclude variables with a *p*-value > 0.10.

## 3. Results

The mean post-operative UCVA was 0.18 ± 0.25 logMar (ranging from 0.00 to 2.77) and BCVA was 0.09 ± 0.56 logMar (ranging from 0.00 to 2.77) with a significant increase (*p* < 0.01) of both parameters after surgery. Moreover, in order to evaluate the overall safety of the procedures, we compared BCVA before and after surgery, observing an improvement in 1362 patients (97.8%). Aiming to assess the effectiveness of our practice, we compared BCVA before surgery and UCVA after it, observing an improvement in 1191 patients (85.6%).

In the entire patient population, mean time to complete resolution of post-operative oedema was 5.27 ± 8.72 days (ranging from 3 to 90). In 13 patients, topical therapy with corticosteroid and hypertonic solutions solved the persistent corneal oedema, but pseudo-phakic bullous keratopathy occurred in 14 patients (1.006%) and required a second intervention of endo-keratoplasty. 

The incidence of complications was as follows: 42 patients (3.02%) developed PCO that required Nd:YAG laser capsulotomy, in 38 patients ophthalmoscopy and OCT macula scan detected a CMO with an incidence rate of 2.73%, 16 cases (1.15%) presented transient IOP elevation, 10 cases (0.72%) had partial iris atrophy and nine patients (0.65%) had an IOL dislocation which required repositioning surgery. 

At the last follow-up, mean = 6.66 ± 8.76 months (ranging from 3 to 95 months), 26 patients experienced an unsatisfactory visual recovery, with an incidence rate of 1.87%.

In the AI analysis, the complications classified were type “1”, “2”, “3”, “4”, “5”, “6”, “7” and “8”. All results for 122,437 decision trees are shown in Figure 1. 

Because of the limitations related to the discrepancy between the group of patients who had complications and those who did not, only results for five or fewer features were considered for further analysis. Calculations with these limitations did not exceed one day (24 h). To prevent overfitting, the decision tree was pruned. Each decision tree was tested for overall values of accuracy (ACC); ACC is defined as the number of correctly classified cases/the number of all cases, as a percentage. Correct decisions are considered as full compliance of predictions with ground truth. For example, the prediction of complication number “4” had to be consistent with the reality “4”, etc. From an ophthalmological point of view, it was also important to understand which complications were better or less well detected. For this reason, an additional analysis was performed for each type of complication. Best results of ACC (71.5%) were detected when the following features were involved: “ocular disease”, “BCVA”, “CIL”, “Kmax” and “intraoperative complication which occurred during surgery” (Table 4, Table 5 and Table 6). 

On the other hand, “Extra devices used during surgery (blue)”, “Ocular diseases (glaucoma)”, “complication which occurred during surgery (Capsular Tear and Extra Capsular extraction)”, “Type of surgeon”, “Patient age”, and “Ocular Disease (Glaucoma)” were identified as significant factors in multivariate logistic regression model (Table 7).

## 4. Discussion

The increase in the mean age of the world population leads us to hypothesize that the volume of cataract surgery, which nowadays is just one of the daily surgical procedures performed worldwide, will increase in the near future [35]. Patients undergoing cataract surgery are becoming ever more demanding, so it is mandatory to improve every aspect of pre-operative, surgical and post-operative care, in order to avoid unsatisfactory results [35].

Many papers have evaluated the features involved in the intraoperative and post-operative insurgence of complications after this type of surgery [7,10,36,37,38,39,40,41], but often they have focused on only one or two of them. In contrast, this study has included all the possible complications. Moreover, for the first time, AI has been applied to evaluate the factors associated with post-operative issues.

According to the results observed in this study, the most important preoperative characteristics of eyes which developed one of the listed complications were the presence of concomitant ocular diseases, lower BCVA and higher astigmatism and intraoperative complications.

Hence, patients with these characteristics should be closely monitored during follow up, to detect and manage eventual complications. It is very important to communicate effectively with these patients, not least to reduce risks of loss of adherence to the prescribed therapy. The relationship of trust between physicians and patients has always been a key factor in successful medical practice, and it is important to try to strengthen it further in complicated cases. Given the systematic increase in daily visits and surgery, it is becoming harder to find sufficient time to accurately explain possible complications and the options in order to solve them. Nevertheless, it is vital to find this time.

The results obtained in this study regarding the features associated with complications during or after cataract surgery agree with those provided by Gonzales et al. [36]. The authors observed that poor preoperative visual acuity, intra-operative complications and high complexity are the factors connected with lower visual and functional outcomes. However, this study is multi-centered and has a very short follow up of only 6 weeks compared to that of Gonzales et al. [36].

Both Jacobsen et al. [37] and Oliveira-Ferreira et al. [38] evaluated the impact of the surgical experience on the rate of complications, observing an association between the procedures performed by residents and the higher number of problems detected after surgery. Their studies provide different results compared to the present one. Firstly, AI was used to overcome all the limitations of previously adopted statistical evaluations. Moreover, the first study included only 128 eyes in the overall analysis [37] and the second evaluated only IOP spike 1 day after surgery [38].

Results observed in this study agree with those provided by papers evaluating only one kind of complication such as IOL dislocation by Durr et al. [40] and CMO insurgence provided by Hollo et al. [14], whereas a slightly higher percentage of PCO was detected compared to that published by Hecht et al. [41].

One of the factors which make it difficult to accurately detect features involved in post-operative complication after cataract surgery is the very high discrepancy between eyes that developed issues and those that did not. It is important to note that, in the sample reported in this study, the safety and effectiveness of the cataract surgery was extremely high. Since the statistical multivariate strategy approach has shown a lack of accuracy in this type of analysis, AI was applied to this evaluation, while also taking into account the rise of its importance in ophthalmic research [42]. 

Evaluating our population with a conditional logistic regression model, it is possible to observe some discrepancies in the results provided by the IA approach. Among the factors significantly associated to the onset of complications after cataracts surgery detected by the regression model, the use of an extra device during surgery, the age of patients and the surgeon in training as operator were not considered significant by the classification tree analysis.

These discrepancies in the analysis, if confirmed by further studies, may provide relevant information for physicians to improve both the scheduling and follow up of their cataract surgery cases. In particular, the age of patients and surgeons in training would not be considered as factors associated with higher rates of developing complications after surgery whereas more attention needs to be paid to cases with concomitant ocular disease, higher corneal curvature, lower BCVA before surgery and the ones that experienced complications during cataract operations.

According to the results observed in this study, AI excludes the influence of systemic diseases such as diabetes or other vasculopathies in post-operative complications. This is a very interesting topic which is much debated [1,40]. However, the role of residents does not appear to be relevant in the onset of this kind of problem, as hypothesized by other studies [37,38]; thus, it is important to emphasize that potentially more complex surgical cases, such as in eyes affected by Pseudo-exfoliation Syndrome or with patients affected by benign prostatic hyperplasia, are not performed by residents in our University. 

One of the limitations of this study, due to its retrospective design, is that few dependent variables have been considered. This is the reason why for instance, phacoemulsification time and power were not included. The time of surgery TOS was included (Table 3) and the results show that it did not affect the insurgence of complications after surgery. This is a very interesting point because it means that a longer procedure, if correctly performed, does not necessarily induce a higher number of complications.

Another limitation of this study may be the value of ACC best results. Although 71.5% is not an extremely high percentage, the authors assume that it is an acceptable limit of accuracy. Further studies, with a prospective design should allow an AI approach to obtain higher ACC levels.

One of the most important advantages of the application of AI is the possibility of building customized predictive models which can be applied in various clinical scenarios where patient typology, clinical setting or type of physicians involved may be very different. Moreover, these models can be changed over time if differences regarding the local population or the surgeons are detected. 

## 5. Conclusions

This is the first study providing an evaluation of the overall complications detectable after cataract surgery which is not limited to one or few of them, and which applies an innovative approach such as AI. 

In conclusion, even if these data need to be confirmed by further studies with a larger population and longer follow up, the classification tree suggests that eyes with ocular co-morbidity, lower BCVA, higher astigmatism and those which experienced intra-operative complications develop post-operative complications more frequently. Thus, an additional attention and a different follow up should be planned for these patients, aiming at early detection and better management of complications, thereby avoiding or reducing disappointing results. 

## Figures and Tables

**Figure 1 jcm-10-05399-f001:**
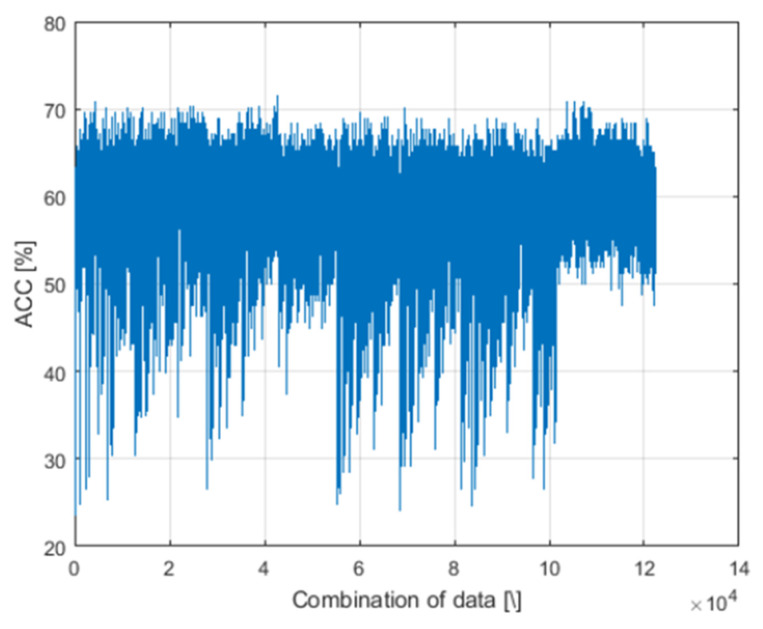
Decision tree of Artificial Intelligence analysis showing the overall values of accuracy (ACC).

**Table 1 jcm-10-05399-t001:** Means, standard deviation (SD) and ranges of ocular preoperative features: uncorrected visual acuity (UCVA) and best corrected visual acuity (BCVA) in LogMar, spherical equivalent (SE) in diopters (D), intraocular pressure (IOP), axial length, mean keratometry, intraocular lens (IOL) power planned to implant, endothelial cell count and anterior chamber depth (ACD).

Parameter	Mean ± SD	Range
UCVA	1.00 ± 0.47	0.05 to 2.77
BCVA	0.76 ± 0.52	0.02 to 2.77
SE (D)	−1.36 ± 4.20	−30 to +8.50
IOP (mmHg)	15.27 ± 2.20	8 to 24
Axial length (mm)	23.87 ± 2.06	17.00 to 35.82
Mean keratometry (D)	44.15 ± 1.74	33.24 to 52.12
IOL power planned to implant (D)	+20.24 ± 5.50	−7.0 to +34
Endothelial cell count (cell/mm^2^)	2400.60 ± 296.27	1293 to 3213
ACD (mm)	3.07 ± 0.44	2.23 to 4.03

**Table 2 jcm-10-05399-t002:** Number and percentage of the most common ocular and systemic concomitant diseases.

Type of Disease	Parameter	Number	Percentage
Systemic diseases	None	272	19.54%
	Hypertension	623	44.76%
	Abnormal heart condition	276	19.83%
	Respiratory disease	160	11.49%
	Mellitus diabetes	251	18.03%
	Benign prostatic hyperplasia	91	6.54%
	Previous cerebral ictus	64	4.60%
	Other	144	10.34%
Ocular disease	None	770	55.32%
	Maculopathy	232	16.67%
	Glaucoma	152	10.92%
	Corneal dystrophy/degeneration	113	8.12%
	Proliferative diabetic retinopathy	62	4.45%
	Pseudo-exfoliation syndrome	25	1.8%
	Amblyopia	21	1.51%
	Previous retinal detachment	12	0.86%

**Table 3 jcm-10-05399-t003:** Results of the preliminary comparative analysis of various methods of artificial intelligence.

Methods	Neural Network	Support Vector Machine	Decision Trees	Naive Bayes Classifier
ACC	68%	69%	71%	60%

**Table 4 jcm-10-05399-t004:** The table below shows the top 10 cases and the different combinations of features and ACC. “0” feature is absent, “1” feature is present. Abbreviations: Eye: laterality of the eye; SD: systemic diseases; OD ocular diseases; CT: cataract type (total; cortico-nuclear; sub-capsular and cortico-nuclear plus sub-capsular); BCVA: best correct visual acuity; SP: spherical defect; CIL: cylinder defect; SE: spherical equivalent; IOP: intraocular pressure; IOL power: intraocular lens power to implant; AL: axial length; Kmax: steeper keratometry; Kmin: flatter keratometry; MK: mean keratometry; ACD: anterior chamber depth measured from the epithelium; ECC: endothelial cell count; TOS: time of surgery; ED: extra devices used during surgery; Intraoperative Complications: complications which occurred during surgery, and Surgeon: type of surgeon (trained or in training).

Features	Different Combinations (0/1 = Present/Absent)									
Surgeon	0	0	0	0	0	0	0	0	0	0
Intraoperative Complication	1	1	0	1	1	0	1	1	1	1
ED	0	0	0	0	0	0	0	0	0	0
TOS	0	0	1	0	0	1	0	0	1	1
Anesthesia	0	0	0	0	0	0	1	0	0	0
ECC	0	0	1	0	0	0	0	0	1	0
ACD	0	0	0	0	0	0	0	1	0	0
MK	0	1	1	1	1	1	1	1	1	1
Kmin	0	0	0	0	0	0	0	0	0	0
Kmax	1	1	0	0	0	0	0	0	0	0
AL	0	1	0	0	0	1	0	0	0	0
IOL power	0	1	0	1	1	1	1	1	0	0
IOP	0	0	1	0	0	1	0	0	0	0
SE	0	0	0	1	0	0	1	0	0	0
CIL	1	0	0	0	0	0	0	1	0	0
SP	0	0	0	0	1	0	0	0	0	0
BCVA	1	0	0	0	0	0	0	0	1	1
CT	0	0	0	0	0	0	0	0	0	0
OD	1	0	0	0	0	0	0	0	0	0
SD	0	0	0	0	0	0	0	0	0	0
Eye	0	0	0	0	0	0	0	0	0	0
Sex	0	0	0	0	0	0	0	0	0	0
Age	0	0	1	1	1	0	0	0	0	0
ACC	71.5	70.8	70.8	70.8	70.8	70.2	70.2	70.2	70.2	70.2

**Table 5 jcm-10-05399-t005:** The effectiveness of the classifier (decision trees) for the 10 best configurations of features broken down into individual complication entities “1”–“8” and patients without complications “0”. The entities are the following: no complications (NC), posterior capsular opacification (PCO), clinical macular oedema (CMO), long lasting corneal oedema solved without surgery (CE), pseudo-phakic bullous keratopathy (PBK), transient elevated intraocular pressure (EIOP), partial iris atrophy (PIA) intraocular lens decentration and/or dislocation (LD) and unsatisfactory visual recovery after cataract surgery (UVR).

Best Results	Individual Disease Entities								
	NC	PCO	CMO	CE	PBK	EIOP	PIA	LD	UVR
	Effectiveness (%)								
1	92.30	86.11	84.84	33.33	62.50	57.14	50.00	22.22	73.07
2	100.00	88.88	75.75	44.44	75.00	42.85	70.00	22.22	65.38
3	53.84	83.33	75.75	44.44	62.50	71.42	50.00	44.44	84.61
4	100.00	80.55	78.78	22.22	62.50	64.28	80.00	66.66	53.84
5	100.00	80.55	78.78	22.22	50.00	64.28	80.00	66.66	57.69
6	61.53	88.88	81.81	22.22	87.50	50.00	60.00	22.22	76.92
7	100.00	83.33	84.84	33.33	50.00	64.28	70.00	33.33	53.84
8	92.30	88.88	66.66	55.55	50.00	64.28	80.00	33.33	61.53
9	92.30	80.55	81.81	33.33	0	57.14	90.00	22.22	80.76
10	92.30	69.44	90.90	22.22	37.50	57.14	90.00	22.22	76.92

**Table 6 jcm-10-05399-t006:** Efficiency for the top 10 results and standard deviation of the mean. The entities are the following: no complications (NC), posterior capsular opacification (PCO), clinical macular oedema (CMO), long lasting corneal oedema solved without surgery (CE), pseudo-phakic bullous keratopathy (PBK), transient elevated intraocular pressure (EIOP), partial iris atrophy (PIA) intraocular lens decentration and/or dislocation (LD) and unsatisfactory visual recovery after cataract surgery (UVR).

**Individual Disease Entities**								
NC	PCO	CMO	CE	PBK	EIOP	PIA	LD	UVR
**Efficiency for the top 10 results and standard deviation of the mean**								
88.4 ± 16.7	83.0 ± 5.9	80.0 ± 6.5	33.3 ± 11.7	53.7 ± 23.6	59.2 ± 8.2	72.0 ± 14.7	35.5 ± 17.9	68.4 ± 11.4

**Table 7 jcm-10-05399-t007:** Results of the multivariate logistic regression analysis.

Features	Beta	Standard Error	*p*-Value	Exp(Beta)	95% Confidence Interval	
					Lower	Upper
Extra device (blue)	−2.001	0.794	0.012	0.135	0.029	0.641
Complication (capsular tear)	−3.125	1.282	0.015	0.044	0.004	0.542
Complication (extra capsular extraction)	−3.73	1.579	0.018	0.024	0.001	0.53
Type of Surgeon (Resident)	0.651	0.288	0.024	1.918	1.091	3.371
Age	0.027	0.012	0.021	1.027	1.004	1.051
Ocular disease (glaucoma)	−1.469	0.652	0.024	0.23	0.064	0.827

## Data Availability

Data presented in this manuscript are available from the corresponding authors on reasonable request.

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
