# Peer review of "Classification Tree to Analyze Factors Connected with Post Operative Complications of Cataract Surgery in a Teaching Hospital"

_jcm, 2021, doi:10.3390/jcm10225399_

Round 1

Reviewer 1 Report

The topic of research seems very interesting and relevant for the journal.

It has not been explained why classification tree is chosen over other methods. Other AI methods should be used as control or for comparison and for proving the efficacy of this classification tree method in this context.

Lot of common English grammatical mistakes and wrong sentence construction. The flow of ideas are not well connected.

Should be re-written with the help of native English speakers.

Author Response

The topic of research seems very interesting and relevant for the journal.

Thank you for the kind words and the time spent in evaluating our manuscript.

It has not been explained why classification tree is chosen over other methods. Other AI methods should be used as control or for comparison and for proving the efficacy of this classification tree method in this context.

Thank you for this comment, we added a short paragraph and a table (table 3) explaining the reason for  our choice and the results obtained with other methods (lines 157-168 and table 3).

Lot of common English grammatical mistakes and wrong sentence construction. The flow of ideas are not well connected.

Should be re-written with the help of native English speakers

Thank you for these comments, the whole manuscript has been edited by a scientific native English speaker aiming to improve the overall quality of it.

Reviewer 2 Report

The authors attempted to use AI classification tree in the evaluation of both ocular and systemic features involved in the onset of complications after cataract surgery in teaching hospital. It was interesting, but there are several questions to be answered and discussed.

First of all, I wonder if the outcomes after analysis by AI classification in this study are quite different from those by conventional analysis such as multivariate logistic regression. The authors compared both analyses together with the same patient data. It was not enough for them to compare the outcomes analyzed by AI with those from other study in the discussion.

In methods, the meaning of spherical defect (SP) and cylinder defect (CIL) need to be explained in detail.

In the table 3, CIL, one of the features was misspelled as CYL.

In results, the authors declared that the best results of ACC was 71.5%. I wonder if 71.5% is quite high accuracy enough to accept.

Author Response

The authors attempted to use AI classification tree in the evaluation of both ocular and systemic features involved in the onset of complications after cataract surgery in teaching hospital. It was interesting, but there are several questions to be answered and discussed.

Thank you for the kind words and the time spent in evaluating our manuscript.

First of all, I wonder if the outcomes after analysis by AI classification in this study are quite different from those by conventional analysis such as multivariate logistic regression. The authors compared both analyses together with the same patient data. It was not enough for them to compare the outcomes analyzed by AI with those from other study in the discussion.

Thank you for this comment, we added  the description of the logistic regression used in the methods section (lines 182-185), the results of it in the results (table 6 and lines 228-232). Moreover, we briefly discussed them in the text ( lines 303-315).

In methods, the meaning of spherical defect (SP) and cylinder defect (CIL) need to be explained in detail.

Thank you for this comment, we better explained the meanings in the methods section (lines 122-124).

In the table 3, CIL, one of the features was misspelled as CYL.

Thank you for this comment, this typing errors has been corrected.

In results, the authors declared that the best results of ACC was 71.5%. I wonder if 71.5% is quite high accuracy enough to accept.

Thank you for this comment, we discussed this in the limitations of the study in the discussion section (lines 331-334).

Round 2

Reviewer 2 Report

I appreciate that the authors have done their best to revise their manuscript as the reviewer's comments and advices.

Author Response

I appreciate that the authors have done their best to revise their
manuscript as the reviewer's comments and advices.

Thank you for your kind comment.